# The Influence of a Seedling Recruitment Strategy and a Clonal Architecture on a Spatial Genetic Structure of a *Salvia brachyodon* (Lamiaceae) Population

**DOI:** 10.3390/plants9070828

**Published:** 2020-06-30

**Authors:** Ivan Radosavljević, Oleg Antonić, Dario Hruševar, Josip Križan, Zlatko Satovic, Doroteja Turković, Zlatko Liber

**Affiliations:** 1Division of Botany, Department of Biology, Faculty of Science, University of Zagreb, HR 10000 Zagreb, Croatia; dario.hrusevar@biol.pmf.hr (D.H.); adajne@gmail.com (D.T.); zlatko.liber@biol.pmf.hr (Z.L.); 2Centre of Excellence for Biodiversity and Molecular Plant Breeding, HR 10000 Zagreb, Croatia; zsatovic@agr.hr; 3Subdepartment of Quantitative Ecology, Department of Biology, Josip Juraj Strossmayer University of Osijek, HR 31000 Osijek, Croatia; oantonic@biologija.unios.hr; 4Multione Ltd., HR 10360 Sesvete, Croatia; jkrizan@multione.hr; 5Department of Seed Science and Technology, Faculty of Agriculture, University of Zagreb, HR 10000 Zagreb, Croatia

**Keywords:** *Salvia brachyodon*, sage, microsatellites, clonality, clonal architecture, seedling establishment, genet, ramet

## Abstract

By performing a high-resolution spatial-genetic analysis of a partially clonal *Salvia brachyodon* population, we elucidated its clonal architecture and seedling recruitment strategy. The sampling of the entire population was based on a 1 × 1 m grid and each sampled individual was genotyped. Population-genetic statistics were combined with geospatial analyses. On the population level, the presence of both sexual and clonal reproduction and repeated seedling recruitment as the prevailing strategy of new genets establishment were confirmed. On the patch level, a phalanx clonal architecture was detected. A significant negative correlation between patches’ sizes and genotypic richness was observed as young plants were not identified within existing patches of large genets but almost exclusively in surrounding areas. The erosion of the genetic variability of older patches is likely caused by the inter-genet competition and resulting selection or by a random die-off of individual genets accompanied by the absence of new seedlings establishment. This study contributes to our understanding of how clonal architecture and seedling recruitment strategies can shape the spatial-genetic structure of a partially clonal population and lays the foundation for the future research of the influence of the population’s clonal organization on its sexual reproduction.

## 1. Introduction

A combination of sexual and asexual reproduction can be found in many plant species [1], and it was estimated that ~80% of angiosperms reproduce by the most common type of asexual reproduction, i.e., vegetative propagation, also referred to as clonal growth [2]. In terms of various biotic and abiotic factors, such as pollinator service, necessity of suitable environmental conditions for flowering, fruiting, seed germination, or seedling establishment, clonality is considered less demanding than sexual reproduction [3,4,5,6].

All individuals who develop through vegetative propagation share the same genotype and are referred to as ramets, while the entire clonal organism that consists of numerous ramets is referred to as a genet [7]. The spatial distribution of ramets (i.e., clonal architecture) defines the levels of clonal aggregation, and two main strategies can be recognized: phalanx and guerrilla [8,9]. A phalanx strategy can be found in species in which clonal offspring remain in close proximity to the parent plant, and such species usually reproduce by tillers, bulbs, or corms [9]. On the other side of the spectrum are species that reproduce by a guerrilla strategy characterized by bulbils, runners, or vegetative propagules as a means of clonal reproduction [9]. In the phalanx strategy, ramets of the same genet are densely aggregated; thus, exclude other genets, while the opposite is observed with the guerrilla strategy, in which the distance among the ramets is considerably longer, resulting in spatial intermingling of genets [9]. As a consequence of the non-random spatial distribution of genotypes, a spatial genetic structure (SGS) emerges [10]. Since the consequence of clonal growth is a spatial distribution of genetically identical individuals in relative proximity to parental plants, SGS is strongly affected by clonal architecture [11].

The level of population genotypic diversity is a consequence of the trade-off between the recruitment of new genets (i.e., successful seedling establishment) and genet die off. New genet recruitment is explained by two main strategies [12,13,14]: initial seedling recruitment (ISR) and repeated seedling recruitment (RSR). In species characterized by the ISR strategy, the recruitment of seedlings usually follows large-scale ecological disturbances. The establishment and development of new genets occurs in short episodes, only through initial colonization but not afterwards [12]. Therefore, it is expected that the number of genets will gradually decrease over time with only a few large genets remaining, resulting in a decrease in the population’s ability to adapt to environmental disturbances [11,13]. In RSR species, seedling establishment can be detected constantly, as new genets establish repeatedly inside the areas occupied by adult genets [12,13,14]. As a result, the number of genets and the level of genotypic variability will remain constant; thus, retaining the population’s ability to cope with environmental changes [11,13,14,15]. As the population grows older, even in RSR species, substantial changes in genet sizes will occur as a consequence of stochastic processes or differences in competition strength among genets [15,16].

*Salvia brachyodon* (Vandas) is a narrowly endemic species restricted to only three known populations in the central Adriatic region [17,18,19]. It was presumed that the species reproduces both sexually through seeding and clonally from perennial underground stolons [20], although no specific research of the species’ reproduction strategy was ever performed. *S. brachyodon* populations comprise of easily recognizable and densely packed patches [20] that also suggest the presence of clonal reproduction. The species is characterized as a hemicryptophyte, and, due to its clonal reproduction, it is likely that resprouting is possible. The studied population is located in the Pelješac peninsula near the mountain top at an altitude from 800 to 860 m.a.s.l. and is the largest of all known populations. Eighteen years before the sampling in this study, a severe wildfire overran the location of the population. The indigenous black pine (*Pinus nigra* (Arnold)) forest that occupied the area before the fire was burned down, and replaced with an open habitat. The evidence of the wildfire is still present, as numerous burned trunks can be observed throughout the *S. brachyodon* population (Figure 1).

In the current study, the spatial-genetic structure of the largest *S. brachyodon* population was thoroughly analyzed. By combining a population-genetic analysis based on a set of microsatellite markers and a detailed analysis of the population’s spatial structure, we sought to answer the following questions: (1) what is the extent of clonality in the studied population, (2) is there any specific pattern in spatial organization of genets in accordance to their size, (3) which types of seedling recruitment and clonal architecture strategies dominate, and (4) what are the influences of seedling recruitment and clonal architecture strategies on shaping the population’s SGS?

## 2. Results

To test the power of each locus set to resolve distinct clones, a genotype accumulation curve was constructed (Figure 2). It revealed that the set of eight analyzed loci was sufficient for the detection of all multilocus genotypes (MLGs) from the analyzed data set.

All probabilities of (1) obtaining a particular genotype under the assumption of Hardy–Weinberg equilibrium (HWE) and by taking into account departures from HWE (*P*_gen_ and *P*_gen_ (*F*_IS_), respectively) and (2) sampling two individuals sharing an MLG derived from different sexual reproductive events under the assumption of HWE and by taking into account departures from HWE (*P*_sex_ and *P*_sex_ (*F*_IS_), respectively), had values of <0.001. Such results confirm that all replicates of the same MLG are most likely part of the same genet and that each distinct MLG belongs to a distinct genet.

To assign multilocus genotypes to multilocus lineages (MLLs), a histogram derived from the pairwise genetic distance matrix (D_psa_) was constructed (Figure 3). However, it was not possible to define the genetic distance threshold that separates MLLs from MLGs and overall, 241 MLGs were identified.

Majority of analyzed loci were characterized by high levels of polymorphic information content (0.56–0.86) while only one loci (SoUZ004) had a moderate value of 0.34. The number of alleles per locus ranged from four (SoUZ004) to 16 (SoUZ006), with an average of 9.12. The results for both observed (*H*_O_) and expected heterozygosity (*H*_E_) were similar, as the obtained average values were 0.72 and 0.70, respectively. Consequently, the inbreeding coefficient *(F*_IS_) had a slightly negative but insignificant value of −0.02 (*p* = 0.75), suggesting that no departure from HWE was present. Obtained values of population genetic parameters are presented in Table 1.

The spatial autocorrelation analysis revealed substantial spatial structuring over short distances of under three metres with the highest absolute values over the first distance classes (Figure 4). With a decomposition of clonal structure, the autocorrelation values sharply decreased. The ramet-level analysis revealed higher levels of genetic structure than the analysis based on the individual MLGs, clearly reflecting the predominance of groups of small clones with closely positioned clone mates over groups of large clones occupying larger areas.

The maximum number of sampled ramets per genet (i.e., clonal size) was 55, while the overall number of samples harboring a unique genotype was 91. In addition, the surface area of the largest genet was estimated at 68 m^2^, while the 91 unique genotype ramets had an estimated surface area of 0.043 m^2^. The genet distribution in accordance with their clonal size and surface area size (*A*_GEN_) is presented in Figure 5.

Genotypic richness (based on the 687 collected ramets sorted into 241 MLGs) had a value of 0.35. The slope of the Pareto distribution had a moderate value of 0.69, while the value obtained for the aggregation index (*A*_c_) was 0.73. A maximum clonal subrange of 10.90 metres was detected in the largest group of patches and refers to the largest genet recognized by 55 sampled ramets. The Shannon–Wiener equitability index had a value of 0.91.

The continuous spatial distribution of genotypic richness yielded by spatial interpolation (Figure 6A and Figure 7A) illustrated a high level of spatial inhomogeneity (areas occupied by one or several genets alternating with those covered by numerous genets).

The geometrical analysis of genet surface area (*A*_GEN_) showed that pairs of genets overlapped in only 3.45% of the total surface area, while overlaps of three or more genets were not detected at all (Figure 6B and Figure 7B). The results strongly indicate that different genets rarely intermingle with each other.

Patch area (*A*_P_) was significantly negatively correlated with patch-level genotypic richness index (*R*_p_: *r* = −0.518; *p* = 0.000), Shannon–Wiener equitability index (*E*_H_: *r* = −0.437; *p* = 0.000) and genotypic spatial mixing index (*S*_mix_: *r* = −0.425; *p* = 0.000), suggesting that an increase in patch area (which also possibly implies an increase in its age) is followed by a decrease in genotypic richness and evenness and the level of spatial mixing within the patch (Figure 8).

Microsatellite scoring data are available in Appendix A.

## 3. Discussion

Microsatellite genotyping of the *S. brachyodon* population from the Pelješac peninsula confirmed its partially clonal structure, as multiple clusters of genetically identical ramets were detected. To avoid the overestimation of levels of the clonal diversity through the identification of possible somatic mutations [21], a one-step threshold was considered. However, it was not possible to define the genetic distance threshold that separates multilocus lineages (MLLs) from multilocus genotypes (MLGs) since no specific gap in the number of pairwise allelic mismatches that is usually present in similar studies [22,23] was detected. If the standard threshold of 2–4% variation [22,24] was applied, this would imply a difference in only one mismatch. Since only three MLG pairs differed by one allele, it was concluded that the rate of somatic mutations was neglectable, if even present. Consequently, we treated these pairs as MLGs and not MLLs. Compared with some other similar studies [22,23], the levels of detected possible somatic mutations were low, likely as a consequence of the rather young age of the majority of the genets. Such an assumption is additionally supported by the fact that the population was dominated by numerous small clones.

The MLG-based population genetic analysis revealed moderately high values off allelic richness, and both observed and expected heterozygosity; thus, suggesting high levels of genetic heterogeneity. Although small and isolated populations are traditionally expected to be characterized by low levels of genetic variability [25], the results from different studies are quite contrasting when compared with each other as some report low levels of genetic variability in studied populations [26,27,28], while some others report the opposite [29,30,31]. Based on the results, the studied population of *S. brachyodon* can be grouped with the populations characterized by high levels of genetic variability.

The spatial autocorrelation analysis on both the genet and ramet levels revealed significantly positive kinship coefficient values (*F*_ij_) for distances under 3 m; thus, indicating a higher than expected genetic relatedness among neighboring individuals and genets. As a consequence of clonal propagation by underground stolons, the spatial genetic structure (SGS) was expectedly more pronounced when all sampled ramets were included. Compared with the ramet-level SGS, the genet-level structure was substantially less pronounced but still significant, likely as a consequence of the barochory-type seed dispersal found in *S. brachyodon* that disable seeds to disperse over longer distances. In addition, the genet-level SGS could also arise as a consequence of pronounced self-pollination [32], but no evidence (i.e., positive inbreeding coefficient value) that could support this assumption on the population genetic level was confirmed.

The overall distribution of clones in accordance with their size (i.e., the predominance of numerous small clones over few large ones) was in the agreement with expectations [33] and was comparable to other similar studies [11,34,35]. The distribution can likely be a consequence of species’ post-fire recovery strategies. In fire-prone ecosystems (e.g., Mediterranean ecosystem), a post-fire recovery of plant species is achieved through different resprouting and seeding strategies [36,37,38]. Although additional evidences that would support the following assumption are needed, it seems possible that *S. brachyodon* shares both of these traits. Since the majority of genets occupy small surface areas, it can be assumed that they emerged after the wildfire. The post-fire recovery of the population likely started through a seedling recruitment from the soil-stored seed bank and was followed by the subsequent establishment of new genets throughout the following years. A survival strategy that relies on depositing the seeds in the soil and waiting for a suitable moment to germinate was well documented in closely related species *Salvia fruticosa* Mill. [38] and many other Mediterranean plant species [39,40,41], and is also expected to be more common in the phalanx-type than in the guerrilla-type species [42]. At the same time, the age of a few largest genets is debatable. Since they are substantially larger than the vast majority of detected genets, it could well be that they survived the wildfire and have afterwards recovered through resprouting which is a known trait of many clonal plant species [43,44,45]. Another possible explanation may be that they also sprouted up after the wildfire from the soil seed bank and in such a scenario, their size can be explained by substantial differences of clonal expansion rates. Such differences can be caused by contrasting genets’ fitness levels [46] or specific environmental conditions at micro-locations of different individuals (e.g., soil depth, canopy closure, soil moisture availability, and sun exposure), which caused uneven growth rates. Such environmental heterogeneity can cause significant differences in resource availability in small spatial and temporal scales, and may result in highly contrasting growth rates of closely positioned individuals of similar age [47,48].

Based on the results, it is possible to discuss the genet recruitment model of the studied population as an important life history trait that has a major influence on the population’s genotypic variability and SGS. If the initial seedling recruitment (ISR) model was dominant at the population level, genets of similar age and presumably of similar size would be expected [12,13]. In addition, the wildfire that overran the location of the population could be considered a strong ecological disturbance after which a single generation of seedlings emerged, and a new population cycle characterized by the predominance of clonal reproduction began. However, the results do not support the ISR model, as the population is characterized by the presence of a few large genets and numerous smaller ones of different sizes, of which the majority is likely only a few years old. Although the supposed initial recovery of the population after the wildfire resembles the ISR model [13], it is clear that gradual establishment of new genets is continuously present; thus, making it obvious that the studied population is characterized by the repeated seedling recruitment (RSR) model. High levels of detected genotypic diversity that are maintained by constant emergence of novel genotypes additionally support this finding, as the ISR model predicts gradual elimination of genotypes and decline of the diversity [11,13].

The moderately high values of the aggregation index (*A*c) supported by the unambiguous results from the geometrical analysis of genet overlapping areas lead to the conclusion that the clonal architecture in *S. brachyodon* population was the consequence of the phalanx strategy. Although the extreme form of the phalanx strategy is achieved through clonal reproduction by bulbs, tillers, or corms [9], clonal reproduction by very short underground stolons detected in *S. brachyodon* is obviously more similar to the typical phalanx than guerrilla strategy. Based on the results, it can be concluded that when two established patches of different genets come into contact through spatial expansion, further expansion of a genet into a territory already occupied by other genets does not occur. In addition, it can be observed that single genets can be occasionally found in several patches of ramets and this is usually the case with larger genets. The fragmentation of genets is likely caused by stochastic environmental factors and habitat heterogeneity, since clonal architecture results do not support the possibility of occurrence of long underground stolons. During the sampling expedition, in several places inside the studied population severe soil disturbances that stretched over several square meters were detected. Likely caused by wild boars, these areas, although surrounded by dense vegetation and numerous *S. brachyodon* patches, were free of any vegetation, suggesting the total destructions of roots and underground stolons. Additionally, the sporadic presence of parasitic plant *Cuscuta* sp. was noticed. Growing on both *S. brachyodon* and locally abundant *S. officinalis*, this parasitic species can easily destroy fractions of larger *S. brachyodon* patches or entire smaller patches and can be also held responsible for their fragmentation (Figure 9).

Furthermore, the detailed spatial genetic analysis revealed a specific pattern in the intra-population spatial distribution of genets in accordance with their size. Small genets, recognized as the sampled ramets characterized by a unique MLG, were almost exclusively found outside the areas occupied by the larger genets, suggesting that developed and densely structured patches of older genets are highly unsuitable locations for new seedlings to establish. Recognized spatial pattern of new genets establishment is likely a consequence of the pronounced phalanx clonal architecture of the population. The significant negative correlations that were found between patch area (*A*_P_) and the individual patch indices *R*_p_, *E*_H_, and *S*_mix_ (genotypic richness, Shannon–Wiener equitability and genotypic spatial mixing index, respectively) clearly illustrate that as the patches become larger and assumingly older, their genotypic richness decreases, likely as a consequence of inter-clonal competition or stochastic die-off of the genets accompanied by the absence of new seedling recruitment within patches. As documented for the closely related species *S. fruticosa* [38], we can assume with great certainty that seedlings of *S. brachyodon* also need clear terrain free of any substantial vegetation cover for successful establishment. Consequently, we can conclude that the RSR strategy, although undoubtedly identified at the population level, cannot be identified at the level of individual patches.

By utilizing such seedling recruitment strategy, the population is oriented toward colonization of the surrounding areas, while the core population comprised of older and larger genets experience a decrease in genotypic diversity due to gradual reduction of the number of genets. At the same time, the large surrounding areas are gradually being occupied by emerging young plants and are consequently characterized by high levels of genotypic diversity. It can be expected that from these “nurseries”, because of inter-genet competition and random genets die off, only a few will survive long enough to reach substantial size. Such observations are in accordance with expectation that natural selection favors the genets with the highest fitness and survival capabilities [33].

Although this research provides substantial insight into the processes that shaped specific SGS of this partially clonal *S. brachyodon* population and shed new light on the spatial development and genetic characterization of its patches, the influence of the population’s clonal structure on its sexual reproduction is yet to be studied. If the species is self-compatible, clonality should positively influence the likelihood of geitonogamous pollination, especially since the population is characterized by the phalanx clonal architecture [49,50]. It is also possible to assume that seeds of outcross origin should be produced in greater numbers in the peripheral parts of large clones while in the central parts, seeds from selfing should be more abundant [51,52]. At the same time, because of the reasons already discussed, it seems likely that from the majority of seeds produced inside large patches, new plants will never establish. Consequently, the phalanx structure could be considered responsible for preventing population genetic degradation through suppressing the establishment of inbred offspring of geitonogamous origin. However, if the species is self-incompatible, clonality could have a negative influence on the sexual reproduction of the population because of pollen discounting [53]. To clarify these issues, thorough research of population’s sexual reproduction and its correlation with detected levels and structure of clonality is needed. The research should be oriented toward the investigation of population’s self-compatibility levels and the influence of population’s SGS and clonal architecture on the sexual reproduction. Only after the comprehensive analysis of the population’s mixed reproduction system, will it be possible to conclude with more certainty whether detected rates of the clonality present a burden for population’s development and stability of genotypic diversity levels, or will the population indeed benefit from asexual reproduction as one the main expansion factors with no negative influence on sexuality and overall genetic variability.

## 4. Materials and Methods

### 4.1. Population Description, Sampling Strategy, DNA Extraction and Genotyping

In the studied population, densely packed ramets form numerous spatially well-defined patches. These patches are a regular round shape when smaller and spatially isolated but are irregular in shape when larger and within clusters of patches. The population is discontinuous, as several spatially well-defined groups of patches were observed (Figure 10).

The entire population was equally sampled. The sampling strategy was based on a rectangular grid with a resolution of one square metre. Since the entire population occupies an area of 385 square metres (i.e., ramets were found in 385 quadrants of 1 × 1 m) (Figure 10), numerous tape measures were used to define the sampling grid in a specific part of the population where the sampling was performed at that time. For each quadrant where plants grew, these sampling criteria were followed: (1) if the species occupied more than 50% of a particular quadrant and only a part of a larger patch was identified within the quadrant, two samples were collected (Figure 11A); (2) if the species occupied less than 50% of a quadrant and only a part of a larger patch was identified within the quadrant, one sample was collected (Figure 11B); and (3) if an individual patch was spatially well-defined and positioned within the quadrant, regardless of the size of the patch and the number of patches within the quadrant, two samples were taken from each patch (Figure 11C,D).

The vast majority of samples were collected following criteria (1) and (2), while only sporadically, and for a very limited number of samples, was criterion (3) applied. With readings at a precision of ~5 cm, each sample was geocoded in an internal coordinate system by measuring its distance from the sides of the quadrant to which it belonged. In addition, all patches were hand drawn at a scale of 1:20 on a millimetre grid, and all the ramets were counted for each particular quadrant. In total, 687 samples were collected and 14,095 ramets were counted. Voucher specimen (ID: 37083) of studied populations has been deposited in the Herbarium Croaticum (ZA) of the Faculty of Science, University of Zagreb.

After sampling, the leaf tissue was desiccated in silica gel, and the genomic DNA was extracted by using a GenElute plant genomic DNA miniprep kit (Sigma-Aldrich, St. Louis, MO, USA).

All individuals were genotyped at eight microsatellite loci (SoUZ001, SoUZ002, SoUZ004, SoUZ005, SoUZ006, SoUZ007, SoUZ011, and SoUZ014) that were previously developed for closely related species (*Salvia officinalis*) and successfully cross-amplified in *S. brachyodon* [54]. PCR amplification was performed in a total volume of 20 µL containing 10× PCR buffer, 1.5 mM of MgCl_2_, 0.2 mM of each dNTP, 0.075 mM of TAIL FOR primer, 0.2 mM of TAIL REV primer, 0.2 mM of M13 primer, and 0.5 U of Taq HS polymerase (Takara Bio Inc., Shiga, Japan). With a GeneAmp PCR System 9700 (Applied Biosystems, Foster City, CA, USA), a two-step PCR protocol with an initial touchdown cycle was applied under the following cycle conditions: 94 °C for 5 min; five cycles of 45 s at 94 °C, 30 s of annealing, beginning at 60 °C and lowered by 1 °C in each cycle, and 90 s at 72 °C; 25 cycles of 45 s at 94 °C, 30 s at 55 °C, and 90 s at 72 °C; and an 8 min extension step at 72 °C. The amplification products were separated by capillary electrophoresis on an ABI 3730XL analyzer (Applied Biosystems, Foster City, CA, USA). The results were analyzed with GeneMapper 4.0 software (Applied Biosystems, Foster City, CA, USA).

### 4.2. Population Level Data Analysis

The Monte Carlo method, as available in the “RClone” package version 1.0.2 [55] and calculated in R version 3.4.3 [56], was used to ensure that the set of loci provided enough power to discriminate among the clones. The unique genotype probability (*P*_gen_) was estimated for each genotype under the assumption of Hardy–Weinberg equilibrium (HWE) as well as by taking into account departures from HWE (*P*_gen_(*F*_IS_)) [21]. To evaluate the likelihood that two samples sharing a multilocus genotype (MLG) were derived from different sexual reproductive events [21], the probability of clonal identity (*P*_sex_) was calculated. In addition to *P*_sex_, the probability of identity, *P*_sex_ (*F*_IS_), was also estimated to consider possible departures from HWE to obtain a more conservative estimate of *P*_sex_. The parameters were estimated by using a round-robin method, described in Parks and Werth (1993) [57], for the estimation of allelic frequencies.

The RClone package version 1.0.2 was also used for MLG identification. Because of somatic mutations or scoring errors [21], MLGs that only slightly differ from each other may be a part of the same multilocus lineage (MLL) [21]. With the intention of assigning the MLGs to MLLs, a pairwise genetic distance matrix (D_psa_) was obtained using the “adegenet” package version 2.1.1 [58] in R version 3.4.3. The pairwise genetic distances between MLGs were converted into a number of distinct alleles and plotted as a histogram to select a threshold based on the number of pairwise allelic mismatches between MLGs that could reasonably be attributed to somatic mutations or genotyping errors [21].

After identification of the samples that belong to identical MLGs, we removed the replicates from the dataset to analyze population diversity. To calculate the polymorphic information content [59] of each locus, Cervus 3.0.7 software [60] was used. GenAlEx 6.5 [61,62] was used to estimate basic population genetic parameters: the average number of alleles per locus (*N*_AV_), the observed heterozygosity (*H*_O_), the expected heterozygosity (*H*_E_), and the inbreeding coefficient (*F*_IS_) as a measure of the population’s deviation from Hardy–Weinberg equilibrium (HWE).

The clonal structure at the population level was analyzed with standardized methods [21] by using the RClone package version 1.0.2. The maximum clonal size (max *n_g_*), genotypic richness (*R*), Pareto distribution, aggregation index (*A*_c_), and clonal subrange [63] were calculated. The statistical significance of the existence of spatial aggregation of clonemates was tested using 1000 random permutations. Following the methods of Loiselle et al. [64], the spatial autocorrelation was estimated by defining distance classes with the same number of units in each class and by calculating the kinship coefficient (*F*_ij_) for each distance interval. To distinguish the impact of clonality from isolation by distance on the population genetic structure, the analysis was performed at two different levels: ramet level and genet level. For the spatial distance computation, the centroid of each MLG was used. To test the given values of *F*_ij_ of each distance class, spatial locations were randomly permuted among individuals 1000 times.

In addition, the Shannon–Wiener equitability index (*E*_H_) [65] was calculated as a measure of homogeneity (evenness) in numerousness different genotypes, ranging from zero (all ramets belong to the same genet) to one (complete evenness with an equal number of ramets belonging to each particular genet), following the equation:(1)$$E_{H}=(-\sum_{i=1}^{S}p_{i}\ln p_{i})/\ln\;S$$
where *S* is the total number of genets in the particular area (i.e., entire population or particular patch) and *p_i_* is the proportion of *i*th genet relative to *S*.

For the purpose of graphical inspection and a better understanding of the genotypic richness spatial distribution throughout the population, the R version 3.4.3 packages “MBA” [66] and “fields” [67] were used. Mapping and construction of density maps were performed by spatial interpolation of the genotypic richness values [68] around each particular sampled ramet in a circle with a radius of 1 m using the MBA algorithm [69]. Density maps were constructed for the majority of the population (~90% of sampled ramets were included), while the sampled ramets that belonged to the patches that were substantially distant from the three major groups of patches were excluded from this analysis.

For an estimation of the area occupied by an individual genet (*A*_GEN_), the relative coordinates for each particular genet represented by at least three sampled ramets were used for the construction of convex hull polygons (CHPs) [70]. Considering that CHP could not be constructed for more than 70% of the genets (i.e., genets with one or two sampled ramets), the average CHP area of genets with three collected ramets was used (dividing by 3, or by 1.5, for genets represented by one or two ramets, respectively) for the average estimation of the *A*_GEN_ missing values. The levels of spatial overlapping were estimated from the CHPs for genets with at least three collected ramets and from buffer polygons (sized to preserve a constant area) for genets with one (circles with constant radius) or two collected ramets (polygons with a variable buffer width with respect to the distance between two ramets belonging to the same genet).

### 4.3. Patch Level Data Analysis

A spatial analysis of genotypic variability at the level of each particular patch was performed in two steps. In the first step, three indices were calculated for each patch: (1) genotypic richness index (*R*_p_, following Dorken and Eckert [68]), (2) Shannon–Wiener equitability index (*E*_H_) (following Equation (1)), and (3) genotypic spatial mixing index (*S*_mix_) calculated as:*S*_mix_ = *N*_Pdif_/*N*_Ptot_(2)
where *N*_Pdif_ represents the number of pairs of the nearest neighboring ramets in the patch that belong to different genets, while *N*_Ptot_ represents the total number of pairs of the nearest neighboring ramets in the patch. To the best of our knowledge, the *S*_mix_ index was used for the first time in this study since no similar expression in the literature was found. Ranging also from zero (all ramets belong to the same genet) to one (each pair of nearest neighboring ramets belong to different genets), this index has an inverse meaning in relation to the aggregation index, *A*_c_ (i.e., a high level of spatial mixing implies a low level of spatial aggregation, and vice versa). The aggregation index is not used for analysis on the patch level because a) it can result in negative values for very small sample sizes and for some specific spatial groups of ramets and genets, and b) *S*_mix_ gives more interpretable results in combination with *R*_p_ and *E*_H_ (i.e., values of all three indices increase with increasing richness, evenness, and the level of spatial mixing of different genets). Consequently, after the first step of this analysis, each patch was described by values of (1) *R*_p,_ which is in proportion to the relative number of genets, (2) *E*_H_, which is in proportion to the evenness of different genets, and (3) *S*_mix_, which is in proportion to the level of spatial mixing of different genets (all ranging from zero to one). In the second step, the values of all three indices were correlated to patch area (*A*_P_; derived from a frame of the internal geodatabase from the digitized hand drawings made in the field) by use of univariate linear regressions.

## Figures and Tables

**Figure 1 plants-09-00828-f001:**
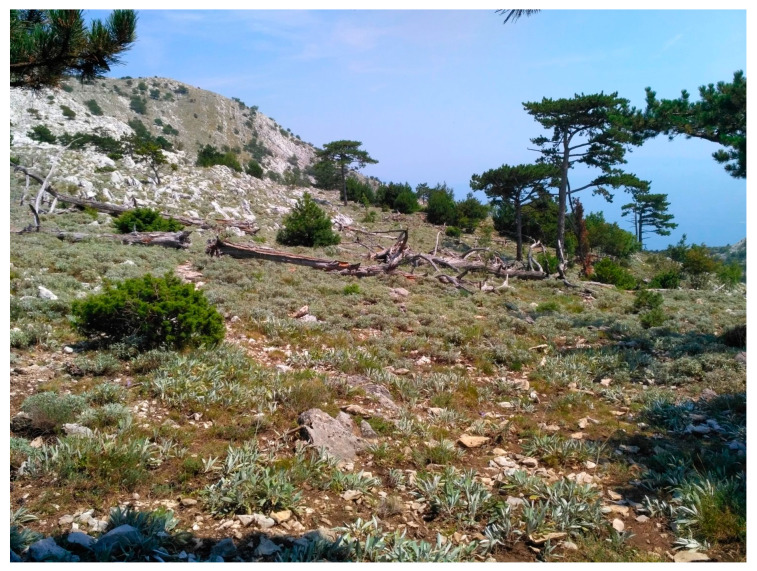
View of the studied *Salvia brachyodon* population. Numerous *S. brachyodon* patches can be seen in the front and burned trunks remaining after the wildfire in the back.

**Figure 2 plants-09-00828-f002:**
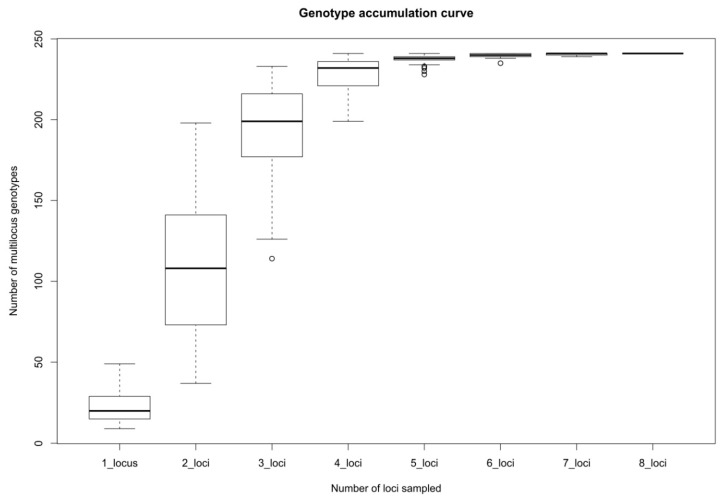
A genotype accumulation curve for 687 sampled individuals of *S. brachyodon*. Box plots were constructed from the number of observed multilocus genotypes (up to 241, the number of unique MLGs in the data set: *y* axis) and the number of assayed SSR loci (*x* axis)

**Figure 3 plants-09-00828-f003:**
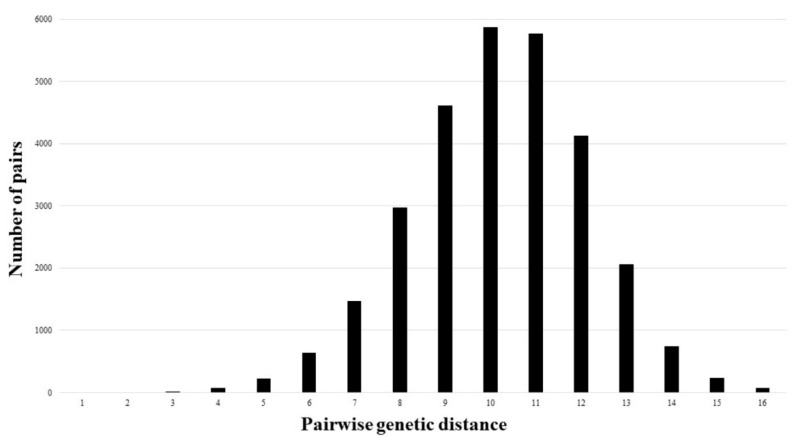
Distribution of pairwise genetic distance frequencies in accordance with given genetic distances.

**Figure 4 plants-09-00828-f004:**
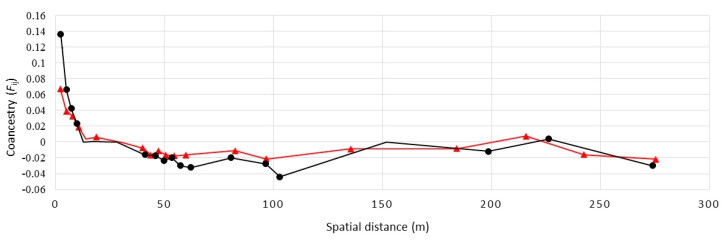
Analysis of the clonal structure of the *S. brachyodon* population by means of a spatial autocorrelation analysis of co-ancestry (*F*_ij_). Genet-level (in red) and ramet-level (in black) analyses were performed. Significant (*p* < 0.05) *F*_ij_ values for a given distance class are marked with red triangles and black circles (for the genet- and ramet-level analysis, respectively).

**Figure 5 plants-09-00828-f005:**
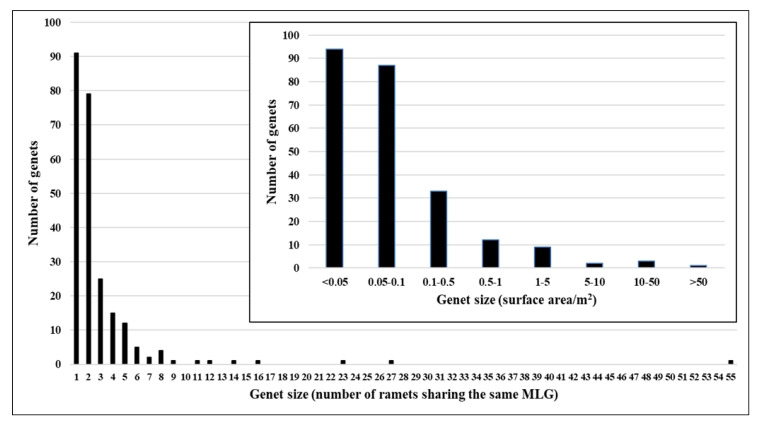
Distribution of genets in accordance with their size: (1) main plot—genet size in terms of the number of sampled ramets sharing the same multilocus genotypes (MLG), (2) small plot—genet size (*A*_GEN_) in terms of the surface areas (m^2^) occupied by the individual genets (represented by convex hull polygons for MLGs with at least three collected ramets and by buffer polygons for MLGs with one or two collected ramets).

**Figure 6 plants-09-00828-f006:**
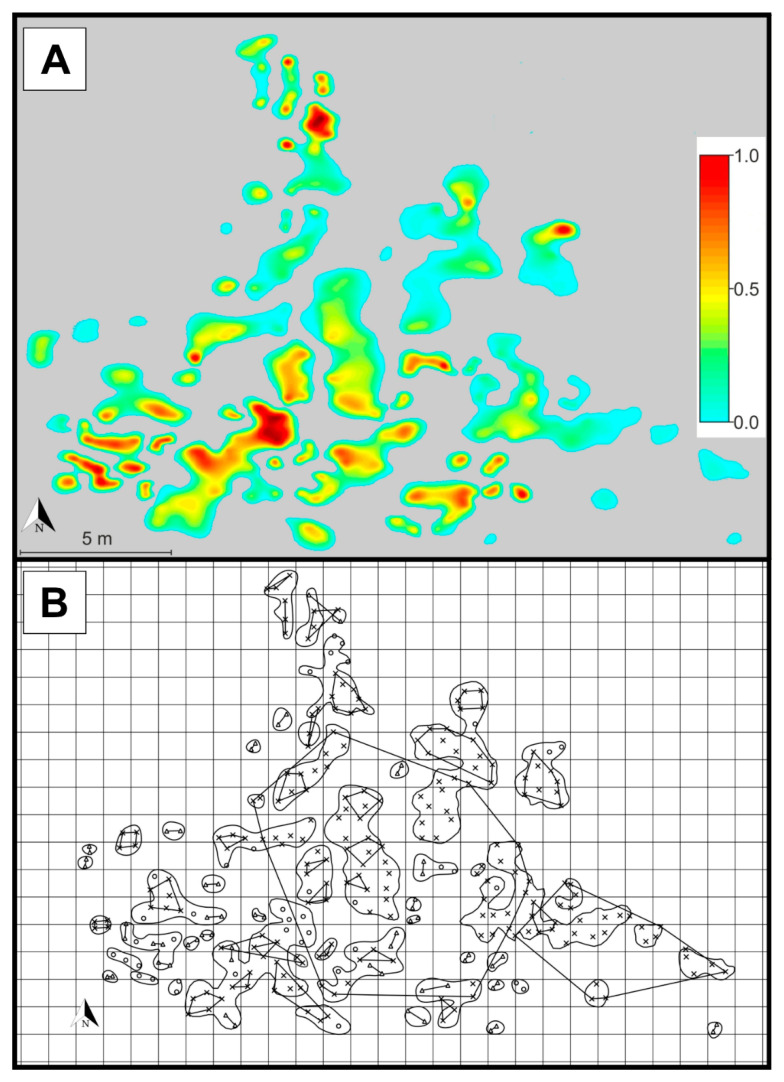
(**A**) Density map showing the spatial distribution of genotypic richness (given by the color scale on the right) in the population’s largest group of patches (eastern group). The patches are colored, while the areas without *S. brachyodon* are in grey. Darker reddish colors indicate areas characterized by high levels of genotypic richness, while lighter greenish colors indicate areas characterized by lower levels of genotypic richness. (**B**) Spatial distribution map of the sampled *S. brachyodon* ramets in the population’s largest group of patches (eastern group) shown over a 1 m^2^ resolution grid. The individual genets that comprise three or more sampled ramets are indicated by convex hull polygons, while the irregular closed shapes indicate the patches of ramets. The sampled ramets that are characterized by a unique MLG are represented by circles, pairs of ramets that belong to the same genet are represented by triangles, and three or more ramets that belong to the same genet are represented by “x” mark and positioned within an associated polygon.

**Figure 7 plants-09-00828-f007:**
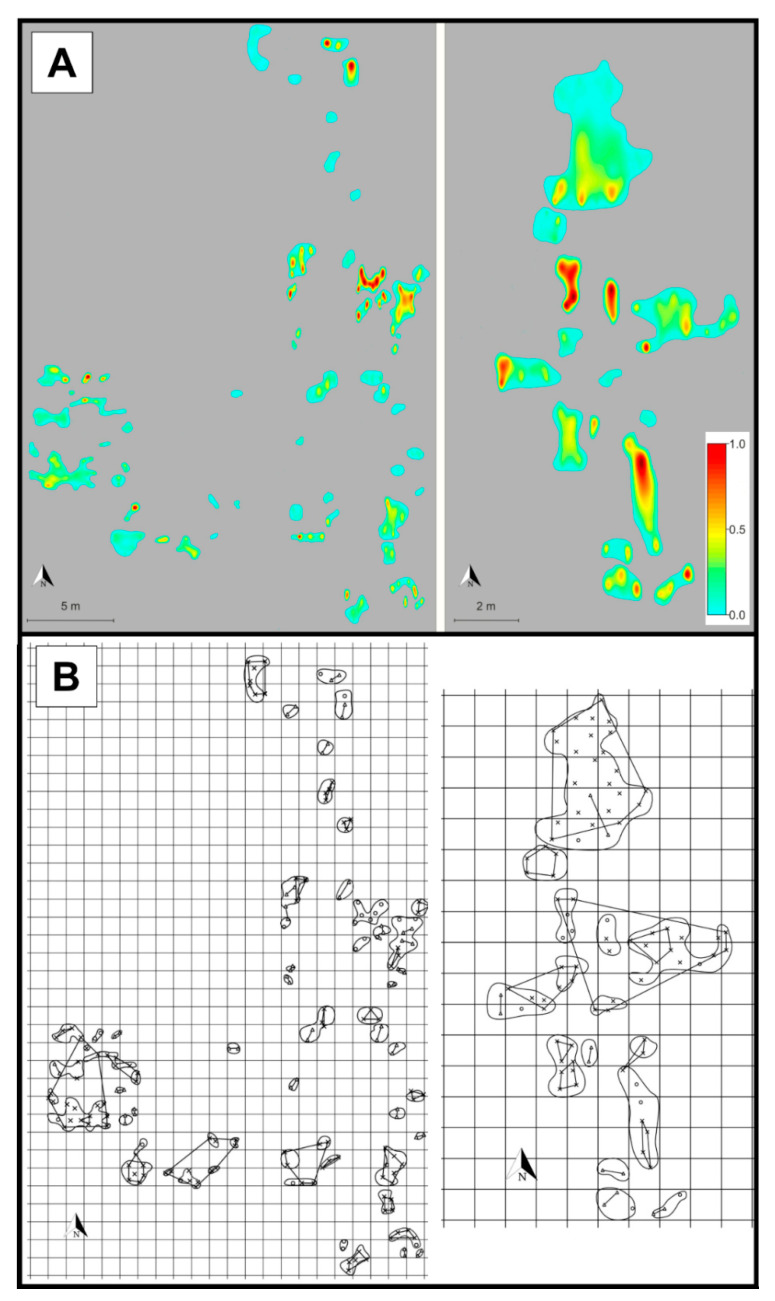
(**A**) Density map showing the spatial distribution of genotypic richness (given by the color scale on the right) in the population’s two remaining large groups of patches (central group on the left and northern group on the right; for better appreciation of the population’s structure, see Figure 10). The patches are colored, while the areas without *S. brachyodon* are in grey. Darker reddish colors indicate areas characterized by high levels of genotypic richness, while lighter greenish colors indicate areas characterized by lower levels of genotypic richness. (**B**) Spatial distribution map of the sampled *S. brachyodon* ramets in the population’s two large remaining groups of patches (central group on the left and northern group on the right: for better appreciation of the population’s structure, see Figure 10) shown over a 1 × 1 m resolution grid. The individual genets that comprise three or more sampled ramets are indicated by polygons, while the irregular closed shapes indicate patches of ramets. The sampled ramets that are characterized by a unique MLG are represented by circles, pairs of ramets that belong to the same genet are represented by triangles, and three or more ramets that belong to the same genet are represented by “x” mark and positioned within an associated polygon.

**Figure 8 plants-09-00828-f008:**
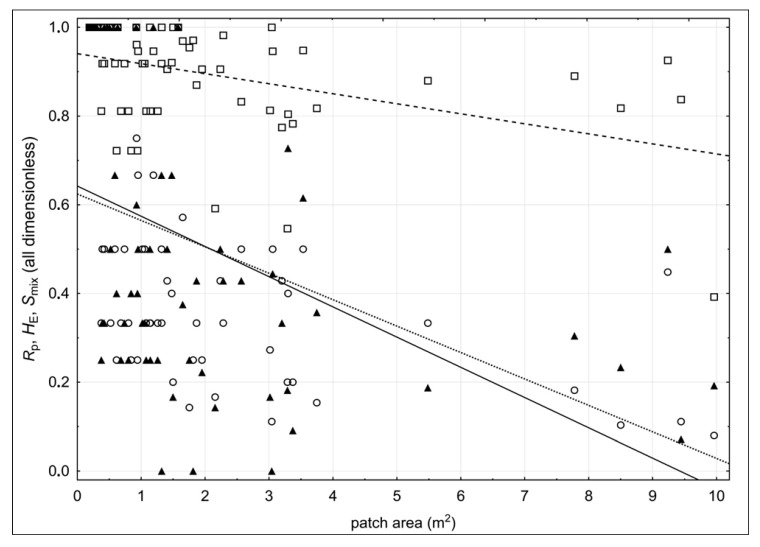
Scatterplots and regression lines of patch area (*A*_P_) against: (1) the genotypic richness index within patch (*R*_p_; circles; solid line), (2) the Shannon–Wiener equitability index within patch (*E*_H_; rectangles, hatched line), and (3) the genotypic spatial mixing index within patch (*S*_mix_; triangles; dotted line).

**Figure 9 plants-09-00828-f009:**
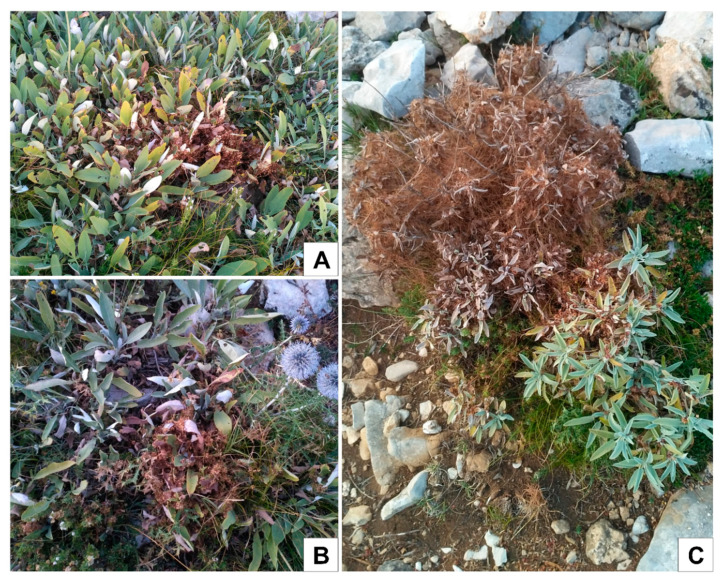
*Cuscuta* sp. on patches of *S. brachyodon* (**A**), and (**B**), and *S. officinalis* (**C**) within studied population.

**Figure 10 plants-09-00828-f010:**
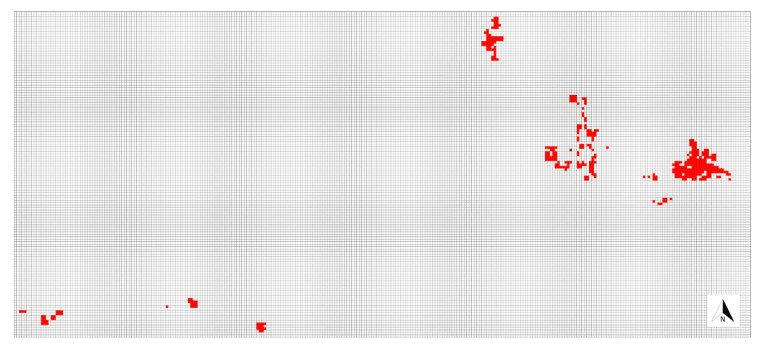
Overview of the spatial distribution of patches in the studied population based on the rectangular grid with a resolution of one square metre. Quadrants of 1 × 1 m where *S. brachyodon* was detected are filled with red color. Three well defined groups of patches can be recognized (northern, central, and eastern) while in the southwest part of the population several smaller groups of patches can be found.

**Figure 11 plants-09-00828-f011:**
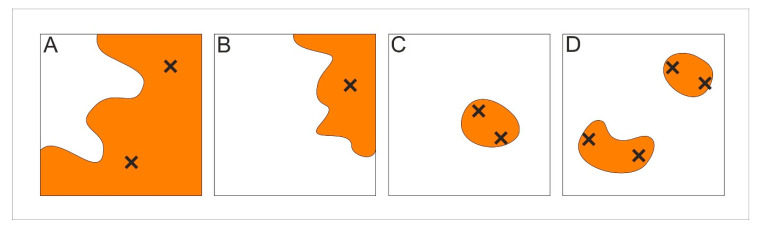
Illustration of the sampling strategy. (**A**), (**B**), (**C**), and (**D**) represent examples of the possible situations in the 1 × 1 m quadrants. The orange fields of irregular shape represent *S. brachyodon* patches, while the “x” marks symbolize the sampled ramets.

**Table 1 plants-09-00828-t001:** Population genetic parameter estimates for each microsatellite locus surveyed in studied *S. brachyodon* population. *N*a, number of alleles; *H*_O_, observed heterozygosity; *H*_E_, expected heterozygosity; PIC, polymorphic information content; and *F*_IS_, inbreeding coefficient.

Locus	*N*a	*H* _O_	*H* _E_	PIC	*F* _IS_
SoUZ001	8	0.643	0.661	0.613	0.026
SoUZ002	5	0.643	0.611	0.562	−0.052
SoUZ004	4	0.369	0.367	0.342	−0.005
SoUZ005	7	0.734	0.739	0.694	0.006
SoUZ006	16	0.867	0.836	0.818	−0.037
SoUZ007	11	0.909	0.877	0.864	−0.036
SoUZ011	14	0.830	0.759	0.724	−0.093
SoUZ014	8	0.751	0.757	0.715	0.007
Mean	9.125	0.718	0.701	0.666	−0.023

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
