# Peer review of "The Influence of a Seedling Recruitment Strategy and a Clonal Architecture on a Spatial Genetic Structure of a Salvia brachyodon (Lamiaceae) Population"

_plants, 2020, doi:10.3390/plants9070828_

Round 1

Reviewer 1 Report

The authors describe their investigation of a Salvia brachyodon population. The genetic structure indicates that a phalanx strategy and not a guerrilla strategy is used by this species. Several hypotheses about the history of the studied population are discussed. Generally, the manuscript is well written. A concise introduction provides the necessary background. Results are clearly presented and well illustrated in multiple figures and supplementary figures. The discussion clearly presents possible limitations of this study and alternative explanations for the presented observations. I cannot see any major flaws in this study and only have a few minor comments and questions.

1) The abstract says that "random die-off" is responsible for the development over time (line 33). This would imply genetic drift. Can the authors exclude that selection (maybe different selection pressures) is ongoing?
2) The results section contains multiple abbreviations which should be introduced at first occurrence.

3) What are the formulas describing regression lines in Figure 4? The authors might want to add these.

4) Are there molecular markers to distinguish phalanx and guerrilla strategy? Are the underlying loci/genes known?
5) Initial seedling recruitment (ISR) (line 239) is explained in introduction and discussion. The authors could explain other abbreviations in the discussion again to be consistent and to make it easier for readers to follow.

6) The material and methods section states that PCR was performed in 20mL reactions. The authors might want to check this.

7) Some additional information about the microsatellite loci could be included here as this is the basis for all analyses. How polymorphic are these markers?

8) It appears that Smix is also a result of this study. The authors might want to present this in the results/discussion section.

9) line 195: in the agreement > in agreement

10) line 283: discussed, I seems > discussed, it seems

11) The authors might want to check the formatting of the supplementary file list.

Author Response

Response to the comments of Reviewer#1

We truly appreciate all the constructive comments and suggestions from the Reviewer. We addressed all comments and requested modifications along with our answers are listed below, written in italic:

The authors describe their investigation of a Salvia brachyodon population. The genetic structure indicates that a phalanx strategy and not a guerrilla strategy is used by this species. Several hypotheses about the history of the studied population are discussed. Generally, the manuscript is well written. A concise introduction provides the necessary background. Results are clearly presented and well illustrated in multiple figures and supplementary figures. The discussion clearly presents possible limitations of this study and alternative explanations for the presented observations. I cannot see any major flaws in this study and only have a few minor comments and questions.

1) The abstract says that "random die-off" is responsible for the development over time (line 33). This would imply genetic drift. Can the authors exclude that selection (maybe different selection pressures) is ongoing?

R: We agree that it is not possible to exclude the selection as an important element in shaping the genetic structure of individual patches and we discussed this in the Discussion section. Therefore, if the Reviewer agree, we suggest rewriting the sentence to: “The erosion of the genetic variability of older patches is likely caused by the inter-genet competition and resulting selection or by a random die-off of individual genets accompanied by the absence of new seedlings establishment.” L: 33-35.

2) The results section contains multiple abbreviations which should be introduced at first occurrence.

R: Indeed. This omission happened when we moved the Materials and methods section from its original position (after the Introduction and before the Results) to the present position. Corrections are now implemented. L: 106, 116, 209-211.

3) What are the formulas describing regression lines in Figure 4? The authors might want to add these.

R: Here are the formulas:

EH = 0.9406-0.0226 x

Rp = 0.6422 – 0.0681 x

Smix = 0.6243 – 0.0596 x

where „x“ is a patch area (AP).

If the review think this is very important, we will add formulas to the manuscript, but we feel that perhaps they are not so important to the readers. We think that the Figure 8 (ex Figure 4) is sufficiently informative and that the main finding visualized by the figure (i.e. as the patches become larger and assumingly older, they will consist of fewer genets) is adequately presented to the readers. At this point, we did not include these formulas into the text, but as already mentioned, if the reviewer think otherwise, we are willing to do so.

4) Are there molecular markers to distinguish phalanx and guerrilla strategy? Are the underlying loci/genes known?

R: To best of our knowledge, such markers do not exist i.e. genes that underlie this trait (i.e. type of clonal architecture) in any species (S. brachyodon or any other) are not known.

5) Initial seedling recruitment (ISR) (line 239) is explained in introduction and discussion. The authors could explain other abbreviations in the discussion again to be consistent and to make it easier for readers to follow.

R: Done. When mentioned for the first time, all abbreviations are now explained throughout the Discussion section: L:227-228, 245, 247, 252, 323-324.

6) The material and methods section states that PCR was performed in 20mL reactions. The authors might want to check this.

R: Yes, of course, it should be µL. The correction is implemented. L. 404.

7) Some additional information about the microsatellite loci could be included here as this is the basis for all analyses. How polymorphic are these markers?

R: We agree. The other Reviewer gave a similar comment. Table 1 is now constructed with additional information regarding each SSR loci. Also, a better explanation regarding the origin of these markers (they were characterized through cross-amplification from Salvia officinalis: Radosavljević et al. 2012) is provided. PIC was calculated for each loci. See lines: 123-129, 401-403, 433-434.

Locus

Na

Ho

He

PIC

FIS

SoUZ001

8

0,643

0,661

0.613

0,026

SoUZ002

5

0,643

0,611

0.562

-0,052

SoUZ004

4

0,369

0,367

0.342

-0,005

SoUZ005

7

0,734

0,739

0.694

0,006

SoUZ006

16

0,867

0,836

0.818

-0,037

SoUZ007

11

0,909

0,877

0.864

-0,036

SoUZ011

14

0,830

0,759

0.724

-0,093

SoUZ014

8

0,751

0,757

0.715

0,007

Mean

9,125

0,718

0,701

0.666

-0,023

Table 1. Population genetic parameter estimates for each microsatellite locus surveyed in studied S. brachyodon population. Na, number of alleles; HO, observed heterozygosity; HE, expected heterozygosity; PIC, polymorphic information content; FIS, inbreeding coefficient

8) It appears that Smix is also a result of this study. The authors might want to present this in the results/discussion section.

If we understand correctly, the reviewer suggest that additional discussion regarding the Smix (the genotypic spatial mixing index) result is needed. However, we used this value in combination with  Rp (patch-level genotypic richness index) and EH (patch-level Shannon-Wiener equitability index) only to test the hypothesis that older/bigger patches are characterized by lower levels of genetic/genotypic variability. We think that the Smix result for itself is not so important to discuss it separately, especially knowing it is in agreement with the results from other two analyses (Rp and EH). Also, the other reviewer suggested simplification of the Discussion section, so we prefer not to add any additional text there. However, if the Reviewer still thinks this is very important aspect of the research, we are willing to consider this suggestion. At this point, we did not implement any changes into the text.    

9) line 195: in the agreement > in agreement

Done.

10) line 283: discussed, I seems > discussed, it seems

Done.

11) The authors might want to check the formatting of the supplementary file list.

The other reviewer suggested that all supplementary figures should become standard figures within the main body of the manuscript, so they are now formatted accordingly.

Reviewer 2 Report

Although I find the subject of this manuscript interesting, some changes are necessary.

I do not understand the inclusion of numerous data as supplementary materials. Most of the Results section is the description of these complementary results. This is atypical because, as the name suggests, supplemental materials should be additional or complementary information, and not the essential part of the study.

The abbreviations used in the text must be explained the first time they are mentioned. Certainly, these are described in the Material and Methods section, which traditionally usually precedes the Results section, but the structure in this journal is different; consequently, authors should describe the abbreviations used in the Results section and not in the Material and Methods section.

The authors indicated that they used eight microsatellites (Table 1) [line 325]; however, there is no table in this manuscript. They probably refer to Table 1 in their previous article [Radosavljević, I.; Satovic, Z.; Jakse, J.; Javornik, B.; Greguraš, D.; Jug-Dujaković, M.; Liber, Z. Development of new microsatellite markers for Salvia officinalis L. and its potential use in conservation-genetic studies of narrow endemic Salvia brachyodon Vandas. Int. J. Mol. Sci. 2012, 13, 12082-12093.], but readers of the current study should have all the information here.

Figures 1 and 3 could be merged for a better understanding.

DNA analysis data is not shown in the manuscript, and most of the results are statistical analyzes of data not shown. This should be changed.

The Discussion section could be shortened; authors should establish a better relationship with the data obtained. In the current version, this section is too conceptual and difficult to understand based on the data shown.

Author Response

Response to the comments of Reviewer#2

We truly appreciate all the constructive comments and suggestions from the Reviewer. We addressed all comments and requested modifications along with our answers are listed below, written in italic:

Although I find the subject of this manuscript interesting, some changes are necessary.

  1. I do not understand the inclusion of numerous data as supplementary materials. Most of the Results section is a description of these complementary results. This is atypical because, as the name suggests, supplemental materials should be additional or complementary information and not an essential part of the study.

R: We agree. Supplementary figures are now transferred to the main part of the manuscript. Regarding the Reviewer’s fifth comment: we agree that the raw microsatellite scoring data should be available, so the excel document containing these data will now be the only supplementary file.

  1. The abbreviations used in the text must be explained the first time they are mentioned. Certainly, these are described in the Material and Methods section, which traditionally usually precedes the Results section, but the structure in this journal is different; consequently, authors should describe the abbreviations used in the Results section and not in the Material and Methods section.

R: Yes, of course. The other reviewer gave the same comment. Corrections are now implemented.

L: 106, 116, 209-211.

  1. The authors indicated that they used eight microsatellites (Table 1) [line 325]; however, there is no table in this manuscript. They probably refer to Table 1 in their previous article [Radosavljević, I.; Satovic, Z.; Jakse, J.; Javornik, B.; Greguraš, D.; Jug-Dujaković, M.; Liber, Z. Development of new microsatellite markers for Salvia officinalis L. and its potential use in conservation-genetic studies of narrow endemic Salvia brachyodon Vandas. Int. J. Mol. Sci. 2012, 13, 12082-12093.], but readers of the current study should have all the information here.

R: We agree. The other Reviewer gave a similar comment. Table 1 is now constructed with additional information regarding each SSR loci. Also, a better explanation regarding the origin of these markers (they were characterized through cross-amplification from Salvia officinalis: Radosavljević et al. 2012) is provided. PIC was calculated for each locus (as requested from other Reviewer). See lines: 123-129, 401-403, 433-434

Locus

Na

Ho

He

PIC

FIS

SoUZ001

8

0,643

0,661

0.613

0,026

SoUZ002

5

0,643

0,611

0.562

-0,052

SoUZ004

4

0,369

0,367

0.342

-0,005

SoUZ005

7

0,734

0,739

0.694

0,006

SoUZ006

16

0,867

0,836

0.818

-0,037

SoUZ007

11

0,909

0,877

0.864

-0,036

SoUZ011

14

0,830

0,759

0.724

-0,093

SoUZ014

8

0,751

0,757

0.715

0,007

Mean

9,125

0,718

0,701

0.666

-0,023

Table 1. Population genetic parameter estimates for each microsatellite locus surveyed in studied S. brachyodon population. Na, number of alleles; HO, observed heterozygosity; HE, expected heterozygosity; PIC, polymorphic information content; FIS, inbreeding coefficient

  1. Figures 1 and 3 could be merged for a better understanding.

R: We agree. We also merged figures S5 and S6.

  1. DNA analysis data is not shown in the manuscript, and most of the results are statistical analyzes of data not shown. This should be changed.

R: We answered this comment earlier. “microsatellite scoring data” XLS file is now included.

  1. The Discussion section could be shortened; authors should establish a better relationship with the data obtained. In the current version, this section is too conceptual and difficult to understand based on the data shown.

R: At this point, we think that any shortening of the Discussion section would negatively influence the quality of the manuscript. However, we greatly appreciate this comment, as we realized that some discrepancies among Materials and Methods, Results, and Discussion sections regarding their organization were indeed present. Now, we arranged analyses, their results and following comments in the same order of appearance, to avoid any confusion when reading the discussion.

The spatial autocorrelation section within the Results is now moved to lines 147-157.

Part of the Discussion section in which the seedling recruitment strategy is being discussed (refers to the distribution of genets in accordance with their size results) is now moved to lines 278-292.

However, if the Reviewer still thinks that some parts of the Discussion section are not necessary and could be easily excluded without reducing the manuscript’s quality, we kindly ask for more specific comments on this issue. We are willing to consider the implementation of such a request.   

Round 2

Reviewer 2 Report

Authors have addressed all the issues and revised the MS satisfactorily. I value positively the effort made by authors for the revision of this manuscript.

Now, I think that this manuscript can be accepted for publication.